# Attosecond-Angstrom free-electron-laser towards the cold beam limit

A. F. Habib [1,2] ✉, G. G. Manahan[1,2], P. Scherkl [1,2,3], T. Heinemann[1,2], A. Sutherland [1,2], R. Altuiri[1,4], B. M. Alotaibi[1,4], M. Litos[5], J. Cary [5,6], T. Raubenheimer [7], E. Hemsing [7], M. J. Hogan [7], J. B. Rosenzweig [8], P. H. Williams [2,9], B. W. J. McNeil[1,2] & B. Hidding[1,2,10] ✉

Electron beam quality is paramount for X-ray pulse production in free-electron-lasers (FELs). State-of-the-art linear accelerators (linacs) can deliver multi-GeV electron beams with sufficient quality for hard X-ray-FELs, albeit requiring km-scale setups, whereas plasma-based accelerators can produce multi-GeV electron beams on metre-scale distances, and begin to reach beam qualities sufficient for EUV FELs. Here we show, that electron beams from plasma photocathodes many orders of magnitude brighter than state-of-the-art can be generated in plasma wakefield accelerators (PWFAs), and then extracted, captured, transported and injected into undulators without significant quality loss. These ultrabright, sub-femtosecond electron beams can drive hard X-FELs near the cold beam limit to generate coherent X-ray pulses of attosecond-Angstrom class, reaching saturation after only 10 metres of undulator. This plasma-X-FEL opens pathways for advanced photon science capabilities, such as unperturbed observation of electronic motion inside atoms at their natural time and length scale, and towards higher photon energies.

For coherent emission of X-rays, FELs rely on the microbunching of a high-energy, high-quality electron beam in the periodically alternating magnetic field of an undulator. The associated thresholds for the relative electron beam energy spread $\Delta W/W < \rho$, where $\rho$ is the FEL parameter[1], and the normalized emittance $\varepsilon_n < \lambda_r \gamma/4\pi$, where $\lambda_r$ is the target X-ray wavelength and $\gamma$ is the Lorentz factor $\gamma \approx W/m_e c^2 + 1$, are very challenging to meet, in particular for the hard X-ray range at photon energies >5.0 keV. Conventional X-FELs[2] are powered by radio-frequency linac-generated beams that achieve obtainable emittances of the order of $\varepsilon_n \approx 1\,\mu$m-rad for high charge (~1 nC) and $\varepsilon_n \approx 0.1\,\mu$m-rad for a low charge (~10 pC) operation[3], respectively. The emittance in linacs is fundamentally limited, for example, by the thermal emittance at the photocathode gun and coherent synchrotron radiation (CSR) in the required magnetic chicane-based beam compressors, and the electron beams typically have durations of the order of tens of fs. At these emittance levels, the high energies required to reach the hard X-ray range necessitate km-scale machines for acceleration, transport, and X-ray production, and puts limits on obtainable photon pulse characteristics. For example, in order to produce shorter photon pulses, ingenious beam manipulation techniques, such as emittance spoiler foils[4] or density spikes[5,6] have to be used to de-emphasize or promote parts of the electron beam for the X-FEL generation process. There is a trend towards enabling operation at lower charges and associated lower emittances[7] and shorter electron beams.

[1]Department of Physics, Scottish Universities Physics Alliance, University of Strathclyde, Glasgow, UK. [2]The Cockcroft Institute, Daresbury, UK. [3]University Medical Center Hamburg-Eppendorf, University of Hamburg, 20246 Hamburg, Germany. [4]Physics Department, Princess Nourah Bint Abdulrahman University, Riyadh, Kingdom of Saudi Arabia. [5]Department of Physics, Center for Integrated Plasma Studies, University of Colorado, Boulder, CO, USA. [6]Tech-X Corporation, Boulder, USA. [7]SLAC National Accelerator Laboratory, Menlo Park, CA, USA. [8]Department of Physics and Astronomy, University of California Los Angeles, Los Angeles, CA, USA. [9]ASTeC, STFC Daresbury Laboratory, Warrington, UK. [10]Institute for Laser and Plasma Physics, Heinrich Heine University Düsseldorf, Düsseldorf, Germany. ✉e-mail: ahmad.habib@strath.ac.uk; bernhard.hidding@uni-duesseldorf.de

Plasma wakefield accelerators offer three or four orders of magnitude larger accelerating and focusing electric fields, and in turn, can achieve multi-GeV electron beam energies on sub-m-scale distances[8]. They also can produce $\varepsilon_n \approx 1$ μm-rad level beams, but reaching energy spreads sufficiently low for FEL is difficult due to the tens of GV m$^{-1}$-level electric field gradients inside of the 100 μm-scale plasma wave accelerator cavities. In addition, in a plasma wave, electron beams are strongly focused, and in turn exit the plasma accelerator with large divergence. Energy spread then can increase the emittance due to chromatic aberration during extraction from the plasma and in the required subsequent capture and refocusing optics[9–11]. The trinity of longitudinal charge density (i.e. current $I$), longitudinal phase space density (energy spread $\Delta W/W$) and transverse phase space density (emittance $\varepsilon_n$) is amalgamated in the composite key performance parameter 6D brightness $B_{6D} = I/(\varepsilon_{nx}\varepsilon_{ny}0.1\%\Delta W/W)$. Electron beam brightness is a key characteristic for FEL performance[12], and hence improvements in electron beam brightness provided the higher brightness can subsequently also be preserved in the beam transport line, and can potentially be converted into advanced photon pulse production.

Electron beams from plasma wakefield accelerators can today routinely generate incoherent undulator radiation in the visible to soft X-ray spectral range[13–15] but achieving FEL gain is challenging due to the strict electron beam quality and preservation requirements. Ingenious post-plasma compensation approaches have been developed to address individual beam quality limitations, i.e. compensation of energy spread constraints or increasing peak current post-beam generation[16–26], with efforts focused on working towards soft X-ray FEL demonstration[27]. The most advanced experimental success so far has been the recent demonstration of FEL gain in the EUV[28] and IR[29] range and seeded FEL[30]. For even harder FEL photons, current results indicate that improving electron beam quality in terms of 6D brightness is necessary. However, due to the interdependency of the 6D brightness parameters, improvement of emittance, peak current, and energy spread have to be tackled concurrently within the initial plasma stage if one aims to compete with or even exceed the quality of beams in linac-powered X-FELs.

It has been suggested that electron beams from plasma photocathode-equipped PWFA may be able to produce electron beams with lower normalized emittance in the 10s of nm-rad range and hence dramatically higher brightness[31–33]. The PWFA stage itself can be driven either by electron beams from linacs such as in ref. [8], or from compact laser wakefield accelerators (LWFA). Linac-generated driver beams can be produced in a very stable manner, efficiently and with a very high repetition rate, thanks to the decade-long development of linac technology. However, the overall linac-driven PWFA system is then similarly large as the linac. PWFA can be driven by electron beams with much larger energy spread and emittance than what is required for FELs[34], and hence can also be powered by electron beams from LWFA[35–39]. This hybrid LWFA → PWFA is a much more recent approach than linac-driven PWFA, and the stability of electron beams from LWFA is not yet competitive with electron beam stability from linacs. On the other hand, LWFA naturally produces intense, multi-kA electron beam drivers, which is the key requirement for PWFA, and hence enables prospects for truly compact and widespread systems, and additionally offers inherent synchronization between electron and laser beams.

Beyond the generation of beams with sufficient quality, subsequent preservation of normalized beam emittance during extraction and transport from plasma wakefield accelerators is very challenging already at the μm-rad level[40,41]. Hence it is a crucial question whether it is possible to generate and then preserve beams with normalized emittance at the nm-rad level and associated brightness, what fraction of the high-brightness beam may survive transport, and if successfully transported electrons can be harnessed for high-brightness photon pulse generation in undulators.

Here we show with high-fidelity start-to-end-simulations, how to produce sub-fs electron bunches with unprecedented emittance and brightness from a PWFA, how to transport them without loss of charge and under full preservation of emittance at the few nm-rad-level, and how to exploit these ultrabright beams for attosecond–Angstrom level X-FEL pulse production.

## Results

The overall setup is shown in Fig. 1 and consists of three main building blocks: the plasma accelerator stage, where the ultrabright electron bunch is generated, the beam transport stage, where the ultrabright electron beam is captured, isolated and transported, and the undulator, where X-ray pulses are generated.

### Plasma wakefield accelerator and plasma photocathode injector stage

An electron beam with current $I \approx 5.5$ kA and electron energy $W = 2.5$ GeV, which may be produced by conventional linacs[8] or compact laser-plasma-accelerators[37,42] drives the plasma wave (see Figs. 1a–c, 2, and see the section "Methods") in singly pre-ionized helium with a plasma wavelength $\lambda_p \approx 100$ μm in the nonlinear blowout regime, similar to previous PWFA demonstrations[33].

We allow an acclimatization distance of 2.5 cm to support the transverse matching of the driver to the plasma density by balancing space charge, plasma focusing and magnetic pinch forces, to achieve experimentally robust and constant wakefields (Fig. 2b,i). Then, a collinear, sub-mJ-scale plasma photocathode laser focused to a spot size of $w_{0,1} = 5$ μm root-mean-square (r.m.s.), releases ~1.4 pC of electrons via tunnelling ionization of He$^+$ inside the plasma wave (Fig. 1a). This small injector spot size not only minimizes the emittance but at the same time reduces the slice energy spread of the beam[31,32]. These ultracold electrons are rapidly captured and are thereby automatically longitudinally compressed and transversally matched in the plasma wave, and form the ultrabright so-called 'witness' beam with a duration of ~520 attosecond (as) r.m.s. and ultralow projected (average slice) emittance of $\varepsilon_{n,(x,y)} \approx 23$ (17) nm·rad. The witness is phase-locked in the accelerating phase of the PWFA and gains energy at a rate of ~30 GV/m (Fig. 2b,ii), thus reaching 1.75 GeV after ~8 cm of propagation. Crucially, the average slice emittance is preserved on the nm-rad level and projected emittance rises by less than 10 nm·rad mainly due to betatron phase mixing, thus exhibiting projected (average slice) $\varepsilon_{n,(x,y)} \approx 32$ (20) nm·rad.

After ~8 cm, a so-called 'escort' beam is released by a second, stronger plasma photocathode (Fig. 1b) to overload the wake locally in overlap with the witness, in order to reverse its accumulated energy chirp (Fig. 2b,iii). By now, the witness bunch at ~1.75 GeV (Fig. 2c) is immune to space charge forces of the escort charge, and continues to be accelerated without significant (slice) emittance growth (Fig. 1c). Meanwhile, its energy chirp is compensated and the relative slice energy spread adiabatically decreases (Fig. 2c). Indeed, projected and slice energy spread nearly converge, while projected and slice emittance are very similar right from the start. This is not only a remarkable signature of the high beam quality in the plasma but also is a prerequisite to robustly facilitate extraction of the witness beam from the plasma, while preserving its beam quality. The same feature of a small spread of energy and emittance across the beam can then be exploited for the FEL process by enabling global matching of the beam to the undulator instead of slice-by-slice matching.

When approaching optimum energy spread, the plasma density is ramped down (Fig. 2b,iv). The witness beam has higher energy than the driver and escort beam electrons (Fig. 2d), and the spatial overlap by the now transversely expanding escort beam supports a gentle transition into the vacuum. The concomitant low projected (average slice) energy spread of $\Delta W/W \approx 0.08$ (0.04)% and flat longitudinal phase space (Fig. 2e) at the end of the plasma stage is key to nm rad-level

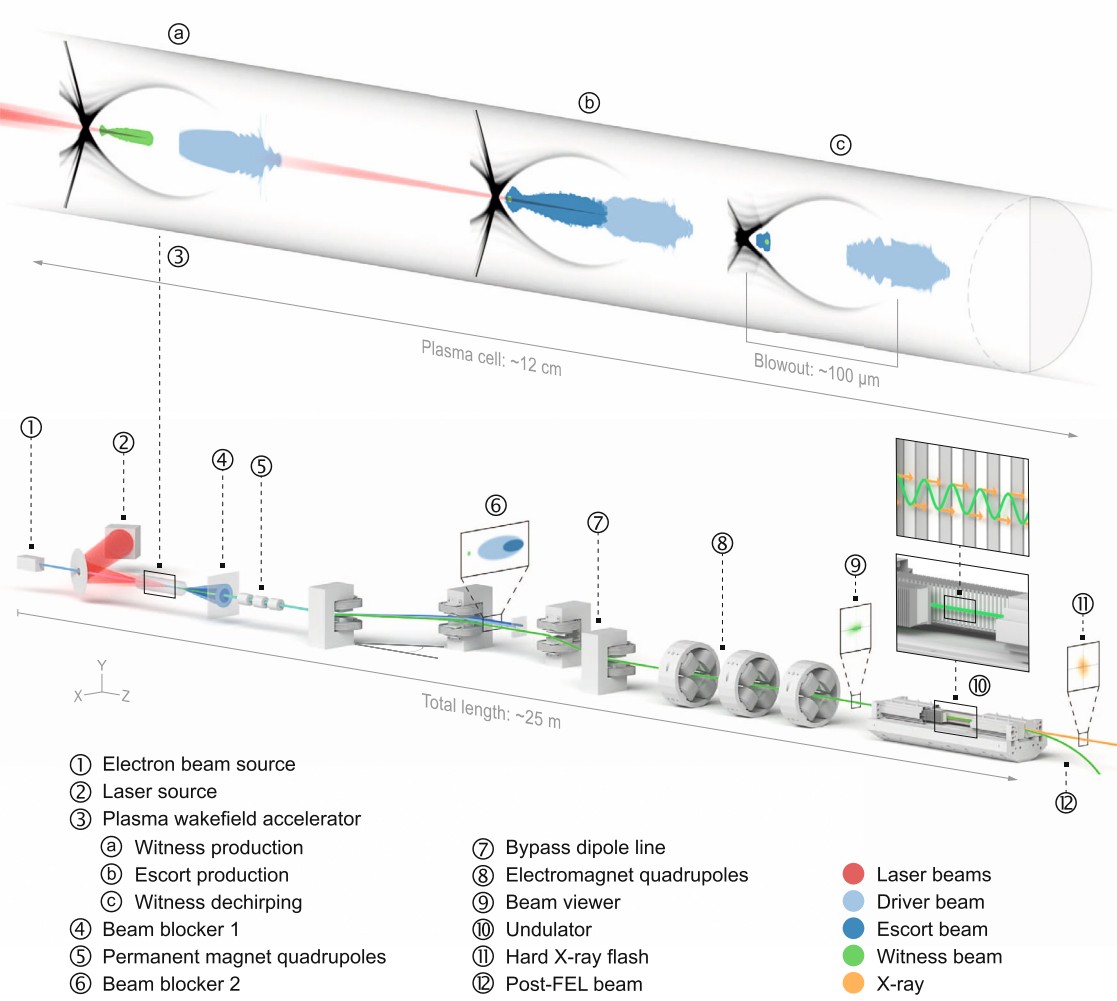

**Fig. 1 | Setup.** The electron beam driver (**1**) excites the PWFA (**3**), where first a collinear plasma photocathode laser (**2**) produces the ultrabright witness electron beam (**a**), and then a second plasma photocathode (**b**) produces a high-charge escort beam that dechirps the witness beam via beamloading (**c**). These three electron populations leave the plasma, and high divergence escort and driver beams are partially dumped into a beam blocker (**4**). The witness beam passes the beam blocker and is captured by a strong permanent magnet quadrupole triplet (**5**). A dipole-based bypass line (chicane) (**7**) dumps (**6**) the remaining driver and escort charge, and the isolated witness bunch (**9**) is matched by an electromagnet quadrupole triplet and focused (**8**) into the undulator (**10**), where X-ray laser pulses (**11**) are generated from the electron witness beam, which is then removed from the axis (**12**).

emittance preservation. While the plasma density ramps down, the associated loss of transverse focusing manifests in increasing beam size of the witness beam, but the low chromaticity enables emission into the vacuum with projected (average slice) emittance of $\varepsilon_{n,(x,y)} \approx 45$ (20) nm-rad, i.e. preservation of slice emittance at the nm-rad level and only ~10 nm-rad projected emittance growth during dechirping and expansion. The witness beam from the amalgamated plasma photocathode injector, PWFA compressor, accelerator, and dechirper reaches projected (average slice) brightness of $B_{6D} \approx 1.3 \times 10^{18}$ ($7.5 \times 10^{18}$) A m$^{-2}$rad$^{-2}$/0.1%bw at the end of the plasma stage. This shows that such a plasma photocathode PWFA stage can act as a brightness transformer and can produce witness beams that are five orders of magnitude brighter than the initial driver beam input (see Supplementary Table 1).

**Witness bunch transport stage**

Driver, escort and witness differ in energy, emittance and divergence, which allows the separation of escort and driver in order to isolate the witness beam (see Fig. 2d, 1 and 3). A permanent quadrupole magnet (PMQ) triplet placed at a drift distance of 10.0 cm from the end of the

plasma is designed to capture the witness beam, which leaves the plasma with increased $\beta$-Twiss of $\beta_{x,y} \approx 0.5$ cm and $\alpha$-Twiss parameter close to zero. The different electron energies allow the separation of the remaining driver and escort electrons from the witness beam in a chicane line. The small bend angle of 2 mrad of the chicane changes the orbit of the high-energy witness beam negligibly, thus avoiding witness beam quality (Fig. 3a) loss by CSR. In contrast, driver and escort electrons are kicked out several millimetres and can thus be dumped, or exploited for diagnostics or other applications (Fig. 3b). The isolated witness bunch passes in its entirety and is then focused into the undulator by an electromagnet (EMQ) triplet. At the entrance to the undulator, the witness bunch has slightly longer r.m.s. duration of $\approx 570$ due to minimal decompression of the bunch tail in the chicane, peak current $I \approx 1.2$ kA, projected (average slice) energy spread $\Delta W/W \approx 0.08$ (0.026)% and projected (average slice) normalized emittance of $\varepsilon_{n,(x,y)} \approx 46.6$ (20) nm-rad (Fig. 3c), and a 6D projected (average slice) brightness of $B_{6D} \approx 1.3 \times 10^{18}$ ($1.1 \times 10^{19}$) A m$^{-2}$ rad$^{-2}$/0.1% bw. These 6D brightness values are far beyond the reach of any other electron accelerator scheme, be it conventionally rf-cavity or plasma wakefield based, and are mutually facilitated by the preservation of

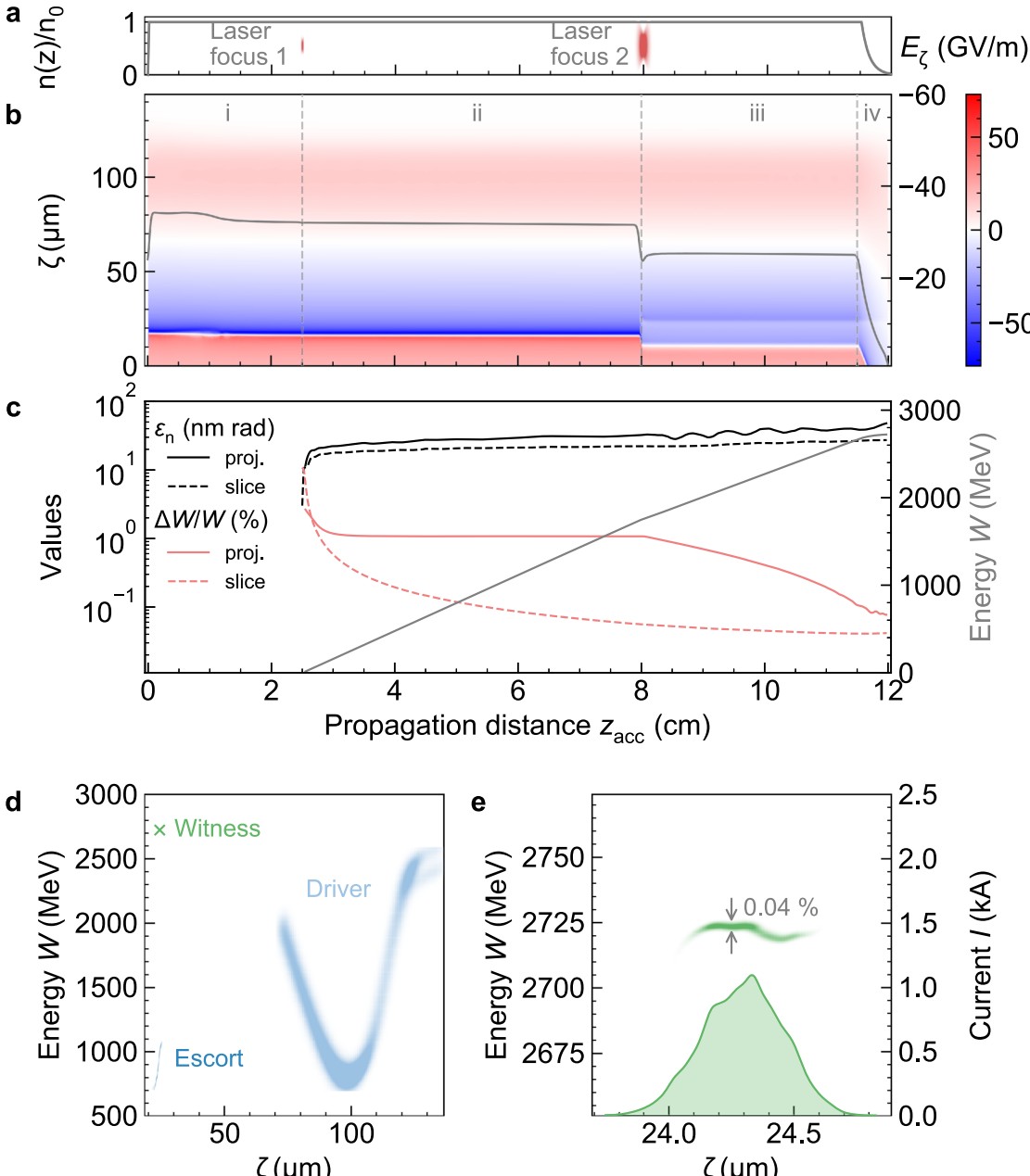

**Fig. 2 | Particle-in-cell simulation of PWFA stage. a** The longitudinal plasma density profile $n(z)/n_0$ is quickly ramped up and remains constant until the extraction downramp. Also shown is the location of the plasma photocathode laser foci that generate witness (**1**) and escort (**2**) electrons. **b** Colour-coded on-axis longitudinal wakefield evolution $E_\zeta$ vs. co-moving coordinate $\zeta$ with the electric field evolution at witness bunch position (solid grey line, right $y$-axis). The dashed lines demark the transition between acclimatization (i), pure witness acceleration (ii), witness dechirping & acceleration (iii), and extraction phases (iv). **c** Projected and average slice emittance and energy spread (left $y$-axis) and energy (right $y$-axis). **d** Overview of longitudinal phase space of electron beams and **e** longitudinal phase space and current of witness beam at the end of the PWFA stage.

emittance (<3 nm-rad average slice emittance $\varepsilon_{n,s}$ growth), and simultaneously of energy spread (even a slight reduction of average slice energy spread $(\Delta W/W)_s$ at the sub-0.1 per mill level) from within the plasma stage through the beamline into the undulator.

The EMQ triplet focuses the witness beam such that at the focal point in the centre of the undulator section the beam size of the "pancake" like witness beam is approximately $\sigma_{y,min} \approx 3.0\,\mu m$ in the undulating plane (see Supplementary Fig. 1). The beam size averaged over the entire undulator length is $\sigma_y \approx 4\,\mu m$ in the undulating plane, and in combination with the ultrahigh brightness and associated gain, no external focusing structures are required to achieve near-ideal overlap of the emerging photon field with the electron beam in one

gain length for optimal coupling. This is a unique configuration in contrast to state-of-the-art X-FELs where typically complex strong focusing schemes are utilized to maintain a small beam size over long undulator sections with corresponding resonance phase correctors.

### X-ray free-electron laser stage

The ultralow emittance can enable coherent X-ray production at ultrashort resonance wavelengths $\lambda_r$ already at the comparatively low energy of only $W \approx 2.725$ GeV, due to the $\varepsilon_{n,s} < \lambda_r \gamma/4\pi$ requirement. We present two X-FEL showcases based on self-amplified spontaneous emission (SASE) in planar undulators of periods $\lambda_u = 5$ mm and $\lambda_u = 3$ mm and peak magnetic fields $B = 2.5$ T and $B = 3.5$ T,

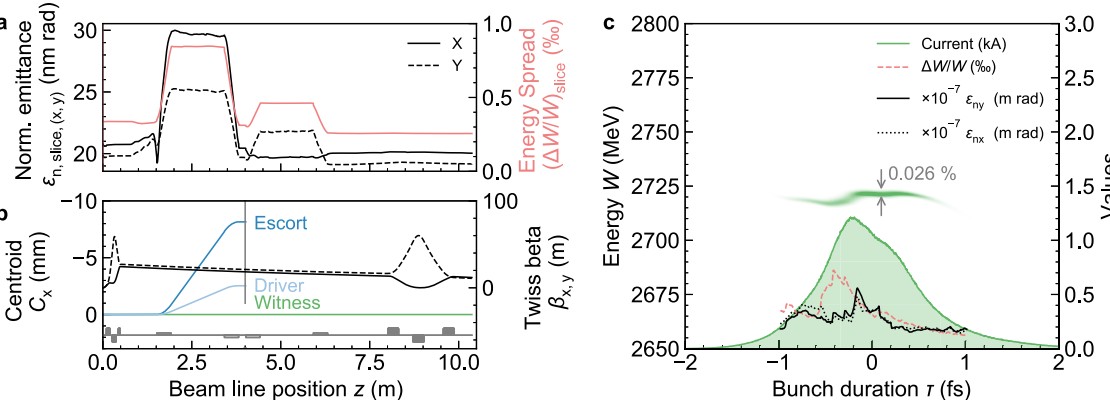

**Fig. 3 | Witness bunch transport. a** Average transverse slice emittance $\varepsilon_{\text{n, slice}}$ evolution and average slice energy spread $(\Delta W/W)_{\text{s}}$ during transport through the permanent magnet triplet, the four chicane line dipoles and the electromagnet triplet. **b** The centroid deflection $C_x$ of the driver, escort and witness (left $y$-axis) and the witness Twiss-parameter $\beta$ ($x$: solid, $y$: dashed) evolution. **c** The witness bunch longitudinal phase space (left $y$-axis) and slice current $I$, energy spread $(\Delta W/W)_{\text{s}}$ and emittances $\varepsilon_{\text{n,s}}$ (right $y$-axis) just before entering the undulator.

corresponding to undulator parameters $K \approx 1.18$ and $K \approx 1$, respectively. With such undulators (see Supplementary Discussion 2 for a discussion on undulator technology), resonance wavelengths $\lambda_r = \lambda_u/(2\gamma^2)[1 + K^2/2] \approx 1.49$ Å and $\lambda_r \approx 0.79$ Å, respectively, can therefore be realized already at the electron energy of $W \approx 2.725$ GeV and a corresponding Lorentz factor $\gamma \approx 5332$.

The 1D FEL coupling parameter $\rho_{\text{1D}}$ can be calculated as $\rho_{\text{1D}} \approx [1/16(I/I_A)(K^2[JJ]^2/(\gamma^3\sigma_y^2 k_u^2)]^{1/3}$, where $I_A = 17$ kA is the Alfvén current, $[JJ] = [J_0(\xi) - J_1(\xi)]$ is the Bessel function factor with $\xi = K^2(4 + 2K^2)$, and $k_u = 2\pi/\lambda_u$ is the undulator wavenumber. For the two examples, this gives $\rho_{\text{1D}} \approx 0.076 \times 10^{-2}$ and $\rho_{\text{1D}} \approx 0.055 \times 10^{-2}$, respectively.

With a relative projected (average slice) energy spread of $(\Delta W/W) \approx 0.08\ (0.026)\%$, theory indicates that the slice energy spread requirement $(\Delta W/W)_{\text{s}} < \rho_{\text{1D}}$ is therefore fulfilled for both cases and due to the low projected energy spread, the resonance condition is maintained within a cooperation length.

The emittance requirement $\varepsilon_{\text{n,s}} < \lambda_r\gamma/4\pi$ is also clearly fulfilled for both cases already at this relatively low electron energy, thanks to the ultralow normalized slice emittance of $\varepsilon_{\text{n,s}} \approx 20$ nm-rad with minimal deviation from the average value along the bunch (see Fig. 3c).

These considerations suggest not only that lasing is possible, but also imply that the corresponding 1D power gain lengths reach the sub-metre range, amounting to $L_{\text{1D}} \approx \lambda_u/(4\pi 3^{1/2}\rho_{\text{1D}}) \approx 30$ cm and $L_{\text{1D}} \approx 25$ cm, respectively. While this represents the idealized case of a cold beam, typically realistic and 3D effects such as energy spread, non-zero emittance and diffraction significantly reduce the gain and hence increase the real gain length. Following the Ming Xie formalism[43,44], one can compute the 3D gain length as $L_{\text{G,th}} = L_{\text{1D}}(1 + \Lambda(X_{\Delta W/W}, X_\varepsilon, X_\text{d}))$, and $\Lambda$ is a function of the scaled energy spread parameter $X_{\Delta W/W} = 4\pi(\Delta W/W)L_{\text{1D}}/\lambda_u$, the scaled emittance parameter $X_\varepsilon = 4\pi L_{\text{1D}}\varepsilon_n/(\gamma\beta\lambda_r)$ and the scaled diffraction parameter $X_\text{d} = L_{\text{1D}}/Z_{\text{R,FEL}}$, where $Z_{\text{R,FEL}}$ is the Rayleigh length $Z_R = 4\pi\sigma_{x,y}^2/\lambda_r$ of the FEL radiation emitted by the electron beam with average slice radius $\sigma_{x,y}$. Since the beam is focused (see Supplementary Fig. 1) to an average beam size $\sigma_y \approx 4.0$ μm in the undulator section, the corresponding Rayleigh lengths of the radiation pulses are $Z_R \approx 1.39$ m and $Z_R \approx 2.55$ m, respectively. Because $Z_R > 2L_{\text{1D}}$, a good overlap between the radiation field and the electron beam is still ensured, gain length degradation due to diffraction is manageable and no further focusing is required. The computed values of the three scaled parameters are $X_{\Delta W/W} \approx 0.198$, $X_\varepsilon \approx 0.048$ and $X_\text{d} \approx 0.479$, and $X_{\Delta W/W} \approx 0.276$, $X_\varepsilon \approx 0.075$ and $X_\text{d} \approx 0.212$, respectively. The ultralow emittance not only manifests its benefit through the short 1D power gain length but on top of that is advantageous in 3D through the minimized pure emittance and shared terms. The 3D power gain length amounts to $L_{\text{G,th}} \approx 49$ cm and to $L_{\text{G,th}} \approx 42$ cm, respectively.

While in current state-of-the-art X-FELs, the difference between 1D and 3D gain lengths is on the level of metres, this small difference between 3D and 1D gain lengths is a signature of an increasingly clean FEL process.

We examine the radiation production with 3D simulations[45] capable of modelling such an X-FEL scenario for the two showcases (see the "Methods" section). In Fig. 4a and d, the resulting power gain is plotted. Thanks to the ultrahigh brightness and high charge density, the SASE process sets in quickly, the lethargy regime ends and exponential gain kicks in already after ~2–3 m. The realistic power gain length can be extracted from the simulation with the best fit at the linear regime and amounts to $L_{\text{G,sim}} \approx 0.54$ m and $L_{\text{G,sim}} \approx 0.62$ m, respectively. This is close to the 3D predictions, and the difference between the observed gain length to the theoretical 1D gain length amounts to only a few tens of cm. The slightly longer gain length of the sub-Angstrom case can be attributed to energy and energy spread diffusion[46,47] and coherent spontaneous emission (CSE) and CSR loss effects. Saturation is reached already after ~10 m, in good agreement with typical estimations of the saturation power length $L_{\text{sat}} \sim 18–20 L_G$.

Final beam powers of (multi-)GW scale are likewise in agreement with the theoretical estimations. At a total electron beam energy of 0.378 J and power $P_b[\text{GW}] = (\gamma m_e c^2/e)[\text{GV}]I[\text{A}] \approx 3.27$ TW, one can use a usual estimation to predict the maximum total radiation power as $P_r \approx 1.6\rho_{\text{1D}}P_b \approx 4$ GW and $\approx 2.8$ GW, respectively. Individual shots come close to this theoretical value; the average radiation powers observed in Puffin are ~4 GW and ~0.5 GW, respectively (see Supplementary Table 2 for a summary of key performance data). In the sub-Angstrom case, the discrepancy between simulated and theoretical values of the saturation power can be attributed to stronger recoil, CSE and CSR losses at shorter radiation wavelengths which may result in reduced coupling between radiation and electron beam.

The radiation propagates faster than the electrons. This inherent slippage of the radiation pulse with respect to the electron beam generally imposes challenges for the longitudinal overlap between the radiation pulse and ultra-short electron beams. For example, the radiation pulse may outrun the electron beam before saturation is reached. For our two showcases the total slippage time is $S = L_{\text{sat}}\lambda_r/(\lambda_u c) \approx 1.1$ fs and $S \approx 0.9$ fs, respectively. This is well within the central electron beam current region, and the ultra-short gain length enables saturation before the radiation pulse outruns the electron beam. As a direct consequence of the ultrashort gain length and slippage length, the cooperation lengths, and the relative electron/light slippage in one gain length, amount to $l_c \lesssim L_{\text{G,sim}}\lambda_r/\lambda_u \approx 16.2$ nm and $l_c \approx 16.5$ nm, respectively. At beam length $\sigma_z \approx 171$ nm r.m.s., the expected number of radiation spikes for the Angstrom and sub-Angstrom case is $M = \sigma_z/$

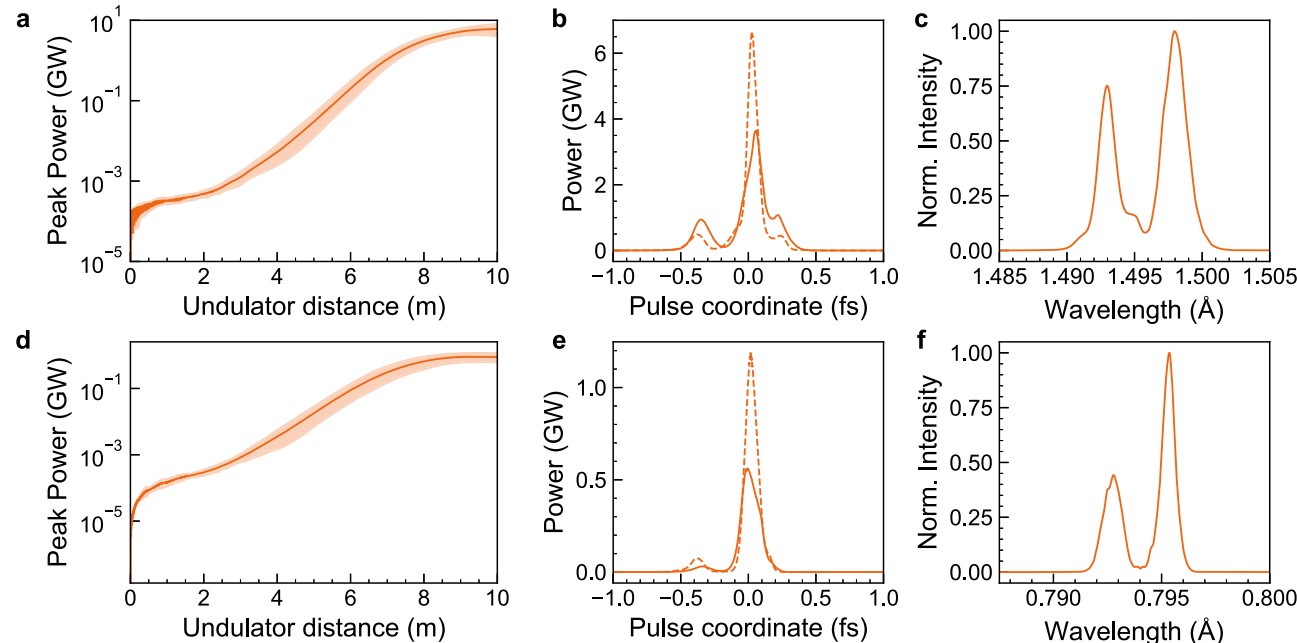

**Fig. 4 | X-ray FEL pulse generation simulated with Puffin.** Panels **a**–**c** and **d**–**f** show power gain over the undulator distance, duration, and spectrum of the produced coherent hard X-ray photon pulses for an undulator with $\lambda_u = 5$ mm (**a**–**c**) and $\lambda_u = 3$ mm (**d**–**f**), respectively. The shaded plot gives the power gain variation for 10 simulated shots with different initial shot noises, and the dark orange solid line is the average power gain across shots. The gain lengths are estimated to $L_{G,sim} \approx 0.54$ m and $L_{G,sim} \approx 0.62$ m, respectively. The solid line in **b** and **e** is the average across 10 shots, while the dashed line is a single-shot representation. In **c** and **f** the radiation spectra are averaged over 10 shots. Figure 1 (inset 11) shows a representative radiation profile at the undulator exit.

$(2\pi l_c) \approx 1.7$ and $M \approx 1.6$, respectively. Therefore, near single-spike radiation pulses are automatically produced without the need, e.g. for delicate beam manipulation methods. This is confirmed by the Puffin simulations shown in Fig. 4b and e. In all shots, an almost entirely isolated, coherent near single-spike pulse is produced, with full width at half maximum (FWHM) average radiation pulse durations of the pronounced spike of approximately $\Delta\tau \approx 100$ as. A concomitant remarkable feature is the clarity of these isolated pulses in the temporal domain, which is an enabling characteristic for true diffraction-before-destruction-type experiments. The variation in radiation power shown in Fig. 4a and d follows the expected statistical properties of the near single-spike regime.

The radiation spectrum shows lasing at the fundamental wavelength for the two respective cases, as shown in Fig. 4c and f. The existence of only two wavelength modes in both cases is the signature of pronounced longitudinal coherence with average FWHM bandwidth of the order of $\Delta\lambda \approx 0.6$ pm around the fundamental wavelength. The corresponding average time-bandwidth product $\Delta\nu\Delta\tau \approx 1.8$ indicates that further improvements may yield Fourier transform limited X-FEL pulses and thus drive the X-FEL to its fundamental limits.

The ultrashort and ultrabright bunches provide an electron beam quality budget that enables the robust realization of tunable X-FEL working points with high contrast in a vast parameter range. Even post-FEL interaction, the beam is still of sufficient quality for demanding further applications (see Supplementary Fig. 1 and Table 1).

## Discussion

While recent experimental plasma-based acceleration results are encouraging with regard to becoming competitive to classical linacs for free-electron-lasing in the EUV and IR range[28,29] and seeded FEL[30], the approach shown here promises to produce electron and photon beams beyond the state-of-the-art even of km-scale hard X-ray facilities. The results from start-to-end simulations of the three components plasma accelerator stage, transport line, and undulator, demonstrate three milestones. First, the synergistic plasma photocathode injector[31,33] and dechirper[32] techniques are shown to be able to produce fully dechirped attosecond electron beams with 0.04% (slice) energy spreads at multi-GeV energies and normalized projected (average slice) emittance of 23 (17) nm-rad. Second, these beams can be isolated and fully transported without significant quality loss in a realistic setup. Contrary to what one may expect in view of the challenges associated with transporting even beams with substantially higher energy spread and μm-rad-level emittance as produced from today's plasma accelerators, the much better initial beam quality achieved directly within the plasma accelerator stage does not aggravate beam quality preservation challenges during transport but instead alleviates beam quality preservation. The final projected (slice) normalized emittance at the entrance of the undulator has grown by only sub-2 nm-rad compared to the plasma stage exit, and the final projected (slice) energy spread remains at nearly initial values (Fig. 3b, Supplementary Fig. 1). Third, these cold beams are predestined for ultrahigh gain and are shown to enable coherent, high-contrast X-FELs when focused into undulators. Distinct coherent hard X-ray pulses with sub-Å, attosecond-scale characteristics are possible without any need for electron beam manipulation or photon pulse cleaning, and further beam and interaction improvements may drive the X-FEL to its Fourier transform limits.

Such attosecond duration hard X-ray pulses constitute advanced capabilities, e.g. for obtaining diffraction images truly unfettered by destruction due to the temporal and spectral purity of the pulses, for imaging of electronic motion on their natural time- and length scale thanks to the temporal characteristics of the pulses at the given photon energy, and for many more applications[48].

This work opens a wide range of applications and configurations, such as prospects of harder photon energies, multi-colour photon pulses, and photon pulses with improved modalities (also see Supplementary Discussion 1). The LWFA-driven PWFA equipped with the plasma photocathode provides a potential pathway towards miniaturization of the technology and may enable using hard X-FELs ubiquitously as diagnostics for probing plasma, nuclear, or high energy

physics and other applications (see Supplementary Discussions 1 and 2). A forward-looking analysis of plasma photocathode stability, repetition rate, efficiency and practical considerations concludes that such a plasma-X-FEL may come within technical reach (see Supplementary Discussions 2–4).

## Methods

### Single-stage plasma wakefield accelerator modelling

We use analytical 1D modelling to design the plasma wakefield setup, and fully explicit, high-fidelity 3D particle-in-cell simulations using the VSim code[49]. The plasma wakefield accelerator stage includes acclimatization, injector, accelerator, dechirper and extractor phases, and is modelled in a single stage. The ultralow witness bunch emittances enabled by our approach are about two orders of magnitude lower than in typical plasma wakefield accelerator simulations. Such emittance and associated brightness values in turn increase demand on the fidelity of the corresponding PIC simulations.

A co-moving window with a Cartesian simulation grid of $N_z \times N_x \times N_y = 1200 \times 120 \times 120 \approx 17.2$ million cells with a cell size of $0.1 \, \mu m$ in the longitudinal direction and $1 \, \mu m$ in the transverse directions is used. We utilized an optimized time step[50,51] corresponding to a temporal resolution of $\Delta t \approx 333$ as. Further, digital smoothing of the currents and VSim's perfect dispersion approach[52] is exploited for the minimization of numerical Cherenkov radiation, and a split field approach is employed for general noise and stray field reduction. Absorption boundaries are used to minimize field reflections.

The driver beam is modelled with variable-weight macroparticles. The background helium plasma is modelled with 1 macroparticle per cell (PPC), while the He$^+$ source medium for the witness beam generation is implemented as cold fluid, which allows adjusting independently the PPC for the witness and escort beam without a significant increase in computational demand. The witness beam here consists of ~200k macroparticles and the escort beam consists of ~1.2 million macroparticles. The much higher number of macroparticles for the escort bunch enables accurate modelling of the dechirping region at the witness beam trapping position. Both plasma photocathode lasers are implemented as Ti:sapphire laser pulses with a central wavelength of 800 nm.

The driver beam is set to a charge of $Q = 600$ pC, the energy of $W = 2.5$ GeV, the energy spread of 2%, and a length of $\sigma_z \approx 12.7 \, \mu m$ r.m.s., corresponding to a current of $I \approx 5.5$ kA, and a projected normalized emittance of $\varepsilon_{n,p,(x,y)} \approx 2$ mm mrad in both transverse planes. The background preionized helium plasma density is $n_0 \approx 1.1 \times 10^{17} \, cm^{-3}$, corresponding to a plasma wavelength of $\lambda_p \approx 100 \, \mu m$. The driver beam is focused to a transverse radius of $\sigma_{x,y} \approx 4.0 \, \mu m$ r.m.s., thus being nearly transversely matched and allowing the PWFA to be driven near-resonantly longitudinally.

The first plasma photocathode tri-Gaussian laser pulse is focused to a spot size of $w_{0,1} = 5 \, \mu m$ r.m.s. and normalized intensity of $a_{0,1} = eE/m_e \omega c = 0.0595$, where $e$ and $m_e$ are the electron charge and mass, and $E$ and $\omega$ are the electric field amplitude and angular frequency, respectively, and a pulse duration of $\tau_1 = 15$ fs FWHM, thus releasing ~1.4 pC of charge from tunnelling ionization of He$^+$. The tunnelling ionization rates are calculated based on an averaged ADK model[53,54]. The second plasma photocathode laser pulse is focused to a spot size of $w_{0,2} = 9 \, \mu m$ r.m.s. and a normalized intensity of $a_{0,2} = 0.062$ with a pulse duration of $\tau_2 = 80$ fs FWHM, thus releasing ~136 pC of charge from tunnelling ionization of He$^+$. Both injector lasers are implemented as envelope pulses in the paraxial approximation. To ensure overlap of the escort beam with the witness, the witness bunch has been released slightly outside the electrostatic potential minimum of the wake in the co-moving frame, whereas the nominal centre of the escort plasma photocathode laser pulse was positioned 15 μm closer to the potential minimum. This shift, in combination with the longer

duration and larger spot size of the escort beam laser pulse, was designed to fully cover the witness beam with the flipped longitudinal electric field for dechirping. The charge yields and focal positions of both laser pulses in the co-moving and laboratory frame, and the corresponding chirping/dechirping rates in the wakefield have been calculated and iterated with a 1D toy model on the basis of ref. [32] to guide exploratory and then production PIC simulation runs.

### Beam transport line modelling

The ~10 m long beam transport line is designed and modelled with the well-known particle tracking code ELEGANT[55]. For a seamless transition from the PWFA stage to the transport line, the full 6D phase space distributions of the driver, escort and witness beam are converted from the VSim PIC code into the ELEGANT 6D phase space distribution format. The beam transport line is optimized for the ~2.7 GeV witness beam energy with the built-in "simplex" algorithm. The whole particle tracking is performed to the accuracy of the third-order transfer matrix. The 6D phase space distributions alongside projected and slice beam properties of all three electron beam populations are individually monitored and analysed. The beam transport line starts with a 10 cm drift distance. In absence of focusing plasma forces, in this section, all three electron populations diverge significantly, but quantitatively very differently, determined by individual beam quality and energy. The driver beam diverges fastest, followed by the escort beam of (from the perspective of plasma photocathodes) moderate quality with a normalized emittance of $\varepsilon_{n,p,(x,y)} \sim 350$ nm-rad, and the high-quality witness beam diffracts least. A collimator with a 0.5 mm aperture just before the first PMQ can filter 40% of the driver charge and a small fraction of the escort beam charge. The collimators are modelled as a black absorbers in ELEGANT. Simplified radiation transport studies of the collimators indicate the feasibility of the filtering approach, enabled by the comparatively low driver and escort electron energies involved; future studies will investigate the technical realization in more detail. The subsequent PMQ triplet is of F–D–F arrangement in $X$-direction (F: focusing, D: defocusing) and opposite in $Y$, where the first two quadrupoles are 10 cm long and the last quadrupole is half that length. The PMQ triplet is optimized such that the witness beam is achromatically collimated at the entrance of the electro-magnet triplet 7.5 m further downstream in the beam transport line. A focusing gradient of the order of 700 T/m is required, which is within reach of today's PMQ technology[56,57]. At projected (average slice) normalized emittance levels of ~46.6 (20) nm-rad, beam quality degradation in terms of emittance growth in the capturing and focusing section of a transport line, even if only of the order of hundreds of nm-rad[11,58], cannot be tolerated. This, driven by the enabling importance of such ultralow emittance and sufficiently low energy spread, is why energy spread reduction has to be secured when still within the plasma accelerator stage. Then, thanks to the low energy spread of the witness beam obtained at the plasma stage exit, beamline achromaticities are inconsequential, and projected and slice emittance, and energy spread, are fully preserved here. After a drift distance of 1 m, the three chromatically discriminative electron populations enter the chicane line, consisting of four rectangular bending magnets (B) arranged in a symmetric C-chicane configuration. We utilized the built-in CSR model to consider beam quality degradation effects[59,60]. Each dipole is $L = 0.4$ m long with a bend angle of $\theta \approx 2.0$ mrad in the horizontal direction. The drift distance between $B_1$–$B_2$ and $B_3$–$B_4$ is $D = 1.5$ m, while the drift distance between $B_2$–$B_3$ is 0.2 m. This results in a relatively small $R_{56} \approx 2\theta^2(D + 2L/3) \approx 0.014$ mm element. After the second dipole $B_2$ the dispersion function of all three populations is at its maximum. However, due to the differences in energy and quality of the three populations, they are deflected by different amounts. The witness beam deviates by a few μm from the design orbit, while the driver and

escort beams are deflected by a few mm in the X-direction. This allows to conveniently isolate the witness beam by inserting a second collimator of 0.4 mm aperture such that the driver and escort beams are blocked and only the witness beam passes through. The last two dipoles B₃–B₄ compensate for the dispersion and the witness beam exits the chicane on the design orbit. It is worthwhile to emphasize that the witness beam is unaffected in terms of beam quality by the CSR effects in the chicane because of the small $R_{56}$-element. After a drift distance of 1.7 m, the witness beam enters the matching section, which consists of an EMQ triplet in an F–D–F arrangement. Each quadrupole is 0.3 m long with a focusing gradient of 45 T/m. The EMQ triplet focuses the witness beam into the undulator section.

### Free-electron laser modelling

The FEL interaction is modelled with the three-dimensional, unaveraged free-electron laser simulation code Puffin[45] in the time-dependent mode. The unaveraged FEL equations allow consideration e.g. of the collective interaction of electrons with broad bandwidth radiation such as electron beam shot-noise, CSE effects and radiation diffraction[61]. Prior to the FEL simulation, the 6D phase space of the witness beam is extracted from the beam transport line simulation and translated into the Puffin format. For accurate FEL interaction modelling, a sufficient number of macroparticles per radiation wavelength is ensured by upsampling the number of macroparticles in 3D with a joint cumulative distribution function from initially ~200k to 3.9 million macroparticles. This produces a smooth current profile on the length scale of the cooperation length and elegantly avoids unphysical CSE emission effects as well. The number of macroparticles in the FEL simulation is approximately half the number of real electrons for the 1.4 pC witness beam. Additionally, we apply a Poisson noise generator on the electron witness beam for realistic shot-noise representation[62].

Following the analytical estimations and numerical modelling, 3D simulations with Puffin are carried through. The magnetic field of the planar undulator with horizontal orientation (see Fig. 1) is modelled in 3D with entrance and exit tapering poles to avoid electron beam steering within the undulator. We integrate the FEL equations with 30 steps per undulator period, in contrast to other undulator period averaging FEL codes. The simulation box is sufficiently large such that it accommodates witness beam evolution and diffraction of the radiation pulse over the entire undulator length. In the longitudinal direction, the electromagnetic field is sampled with 10 cells per resonance wavelength.

We performed 10 simulations for each X-FEL case with initially different shot noises and compute from that the average and standard deviation characteristics of the radiation pulse.

Puffin models the radiation spectrum in its entirety with some wavelength cut-off defined by the user. As such all the wavelengths up to the cut-off are contained within the electromagnetic field of the radiation pulse (see Supplementary Fig. 2). Therefore, we applied a spectral filter to the electromagnetic field to obtain the gain curve, pulse profile, and spectrum of the fundamental mode in Fig. 4.

### Data availability

The data generated in this study have been deposited in the University of Strathclyde publicly available database under accession code https://doi.org/10.15129/176712e5-7677-461e-9d78-bb9af35cff76.

### Code availability

Code and input data associated with the publication are available upon request from the corresponding authors.

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

## Acknowledgements

B.H., P.S., A.S., A.F.H., T.H., and G.G.M. were supported by the European Research Council (ERC) under the European Union's Horizon 2020 research and innovation programme NeXource, ERC Grant Agreement No. 865877. The work was supported by STFC ST/S006214/1 PWFA-FEL and STFC Agreement Number 4163192 Release #3. This work used computational resources of the National Energy Research Scientific Computing Center, which is supported by DOE DE-AC02-05CH11231, and Shaheen (project k1191). The work of E.H. was supported by U.S. Department of Energy Award no. 2021-SLAC-100732. We thank Jonathan Wurtele and Gregory Penn for useful discussions on FEL modelling.

## Author contributions

A.F.H. and B.H. developed the approach and managed its execution. A.F.H. carried through the start-to-end simulations. A.F.H., T.H., P.S., A.S., R.A., J.C. contributed to particle-in-cell simulations; A.F.H., G.G.M., P.H.W., M.L., T.R. contributed to beam transport modelling; A.F.H., B.M.A., B.W.J.McN., contributed to FEL simulations. B.H., A.F.H., M.J.H., T.R., E.H., J.B.R., and B.W.J.McN contributed to the experimental feasibility of the approach. B.H. supervised the project. All authors contributed to manuscript generation.

## Competing interests

A.F.H., G.G.M. and B.H. are inventors of a patent "Plasma Accelerator" WO2018069670A1 (status: published, Applicant: University of Strathclyde, with Radiabeam Technologies SME). The remaining authors declare no competing interests.
