## [Peer Review File · Nature Communications]

Attosecond-Angstrom free-electron-laser towards the cold beam limitEditorial Note: This manuscript has been previously reviewed at another journal that is not operating a transparent peer review scheme. This document only contains reviewer comments and rebuttal letters for versions considered at *Nature Communications*.

REVIEWER COMMENTS

Reviewer #2 (Remarks to the Author):

Dear authors,

Thank you for your thorough and detailed consideration of my comments. The questions are very well answered and clarified in the newly submitted manuscript. In addition, the ambitious parameters in the study are now relaxed to realistic values, which makes the paper very attractive to the scientific community.

I have only one further concern about the presented study to in the supplementary material on the use of higher harmonics and producing radiation with less than Angstrom wavelengths. As I mentioned in my previous letter, in such regimes various effects are intensified which are not typically assumed in FEL equations. I think it makes sense if authors comment on the uncertainty of such predictions. For example, why quantum effects are negligible and why electromagnetic recoil can be disregarded. Some more detailed comments on the accuracy of these numerical results are also valuable. The authors merely mention that the results are very accurate, but I think it is better to write the uncertainty of the results. Since the intensity of the $\lambda/4$ radiation is six orders of magnitude smaller than the main radiation, it raises the question if the observed pick in the plot is really correct. If this value is in the same range as the discretization error, then it is not really a proper basis for showing radiation for the fourth harmonic. In summary, I think some explanation of these important results will add to the quality of the paper.

I think after fulfilling this minor revision, the paper in its present form can be published.

Best regards,

Reviewer #3 (Remarks to the Author):

The authors have updated the manuscript and submitted it to Nature Communications. I am happy to see that they have addressed satisfactorily most of the criticism raised by the three reviewers. The manuscript is now significantly better. I still have, however, the following minor comments:

- The authors mention two times “without quality loss” (line 29 and 330). I think that it is not fully true that there is no quality loss, so I would suggest changing that by “without significant quality loss”.

- Lines 42-43. The authors quote 1 μm emittances at 1 nC and 0.2 μm emittances for 10 pC and cite the work done at SwissFEL (Ref. 3). However, the quoted numbers in Ref. 3 are 200 nm for 200 pC and 100 nm for 10 pC. The authors should correct that.

- When discussing that the PWFA can be driven by conventional linacs or by LWFA, the authors could cite the very recent demonstration on LWFA-PWFA (<https://journals.aps.org/prx/abstract/10.1103/PhysRevX.12.041016>).

- Line 138. I guess that the “23 (20) nm-rad” should be “23 (17) nm-rad”. This would fit with the numbers given in Table 1 of the supplementary material and with the fact that, according to the authors’ response and manuscript, the slice emittance seems to slightly increase from W1 to W2.

- Line 199: “(less than 2 nm-rad average slice emittance (...))”. The authors should check this number (according to Table 1 of the supplemental material it should be 3 nm).

- Lines 291-306 and figure 4. The authors should comment on the fact that the temporal profile has only one mode but the spectrum shows two modes (sorry that I forgot to mention this in the first review).

- Table 1 of the supplementary material. From W3 to W4 there is a slight increase of the pulse duration, so I would expect a proportional decrease of the peak current. However, the current values quoted in the table are the same.

I still think that the main idea of the paper is not new and that the results were somewhat expected from earlier work. Nevertheless, I acknowledge the advances of the authors’ research. They show with

high-fidelity simulations a great brightness improvement compared to previous work, the capacity to transport the electron beam without losing too much of its quality, and the capacity of such electron beams to produce GW sub-fs hard X-ray FEL pulses.

To conclude, I recommend publication in Nature Communications after the authors address the comments previously mentioned.

We would like to thank all Reviewers again for the constructive discussion and feedback, which has improved the manuscript further. Here, we address the remaining points raised.

Please find the corresponding revised manuscript attached and the point-to-point response (our responses in blue) below:

Reviewers' Comments:

Reviewer #2 (Remarks to the Author):

Dear authors,

Thank you for your thorough and detailed consideration of my comments. The questions are very well answered and clarified in the newly submitted manuscript. In addition, the ambitious parameters in the study are now relaxed to realistic values, which makes the paper very attractive to the scientific community.

We thank the Reviewer very much.

I have only one further concern about the presented study to in the supplementary material on the use of higher harmonics and producing radiation with less than Angstrom wavelengths. As I mentioned in my previous letter, in such regimes various effects are intensified which are not typically assumed in FEL equations. I think it makes sense if authors comment on the uncertainty of such predictions. For example, why quantum effects are negligible and why electromagnetic recoil can be disregarded.

We indicated the challenges which may arise from driving the X-FEL towards even harder photon energies in our Supplementary Discussion 1, now lines 152-154: "However, towards even shorter radiation wavelength future studies need to address challenges, such as strong electromagnetic recoil effects, quantized nature of the radiation, and smaller electron density per FEL bucket."

For quantification, the transition from classical to quantum FEL can be identified by the so called "quantum FEL parameter" ρ_{QFEL} presented in Bonifacio, R. et al. Phys. Rev. ST Accel. Beams 9, 090701 (2006) as $\rho_{\text{QFEL}} = \rho_{\text{1D}} m_e c \gamma / \hbar k$. For $\rho_{\text{QFEL}} < 1$ the FEL process would require quantum treatment. Evaluation of this equation for the 4th harmonics of the presented cases, namely 0.32 Angstrom and 0.2 Angstrom, yields $\rho_{\text{QFEL,C1}} \approx 53.4$ and $\rho_{\text{QFEL,C2}} \approx 24.1$, respectively. This indicates that in our showcases we are by a large margin away from the quantum regime of the FEL. That said, we acknowledge the fact that for increasingly shorter radiation wavelengths, the possibility of which we indicate, this will become increasingly imminent and would require quantum treatment of the FEL process. We therefore expanded the discussion and added the following sentence in Supplementary Discussion 1, lines 154-158: "While current results are far away from the quantum FEL regime, according to the "quantum FEL parameter"¹⁰ $\rho_{\text{QFEL}} = \rho_{\text{1D}} m_e c \gamma / \hbar k$ with $\rho_{\text{QFEL,C1}} \approx 53.4$ and $\rho_{\text{QFEL,C2}} \approx 24.1$ even for the 4th harmonics, further push towards harder photons may enter the $\rho_{\text{QFEL}} < 1$ regime and will require quantum treatment of the FEL process."

Some more detailed comments on the accuracy of these numerical results are also valuable. The authors merely mention that the results are very accurate, but I think it is better to write the uncertainty of the results. Since the intensity of the $\lambda/4$ radiation is six orders of magnitude smaller than the main radiation, it raises the question if the observed pick in the plot is really correct. If this value is in the same range as the discretization error, then it is not really a proper basis for showing radiation for the fourth harmonic. In summary, I think some explanation of these important results will add to the quality of the paper.

Indeed, in conventional averaged FEL codes with slowly varying envelope approximation, only radiation frequencies close to the resonance frequency f_r can be modeled accurately. The confidence range of frequencies is limited by the Nyquist condition to $f_r/2 < f < 3 f_r/2$ in these codes, see Campbell, L.T. et. al J. Phys., 16, 103019, (2014). However, these limitations do not exist for unaveraged FEL codes such as Puffin. In contrast, Puffin can model the FEL interaction across a wide range of frequencies simultaneously, limited only by the field sampling rate that defines the Nyquist frequency $f_N = f_s/2$, where f_s is the sampling rate. In terms of wavelength representation, we obtain $\lambda_N = 2\lambda_s$, where $\lambda_s = \lambda_r/10$ is the sampling rate in terms of the resonance wavelength, because we sample the resonance wavelength with 10 cells (see Methods, line 479). This suggests that the shortest radiation wavelength that can be resolved in this configuration is $\lambda = \lambda_r/5$. Therefore, the 4th harmonic $\lambda_r/4$ is well within the numerical accuracy. The capabilities and limitations of the Puffin code are in detail discussed in Ref. [Campbell, L.T. et. al, J. Phys. 16, 103019, (2014)].

We briefly commented on the capabilities of Puffin code in our Methods (lines 459-462) and Supplementary Material (lines 69-73), but did not elaborate more on Puffin's unique features in previous versions of the manuscript, because it is much better covered in literature. However, we agree with the Reviewer that extending this discussion as suggested is appropriate. Therefore, we included the above-mentioned reference in line 462 and extended the discussion in the Supplementary Material which reads now, lines 69-78: "In conventional averaged FEL codes with slowly varying envelope approximation, only spectral modes close to the resonance frequency f_r can be resolved accurately. The Nyquist condition $f_r/2 < f < 3 f_r/2$ provides the confidence range of frequencies in these codes¹. The unaveraged integration of the FEL equations in Puffin enables the capability to model the radiation spectrum over an extended range simultaneously. This allows self-consistent modeling of higher harmonics. The upper frequency limit in Puffin is dictated by the Nyquist frequency $f_N = f_s/2$, where f_s is the sampling rate of the electro-magnetic field. It means that wavelengths down to $\lambda_N = 2\lambda_s$ can be accurately resolved, where λ_s is the discretization cell length. In the present configuration with $\lambda_s = \lambda_r/10$ (see Methods), up to the 5th harmonics $\lambda = \lambda_r/5$ can be computed with confidence."

I think after fulfilling this minor revision, the paper in its present form can be published.

Thank you very much.

Best regards,

Reviewer #3 (Remarks to the Author):

The authors have updated the manuscript and submitted it to Nature Communications. I am happy to see that they have addressed satisfactorily most of the criticism raised by the three reviewers. The manuscript is now significantly better. I still have, however, the following minor comments:

We are likewise glad that we were able to satisfy the criticism, and thank the Reviewer very much for their effort.

- The authors mention two times “without quality loss” (line 29 and 330). I think that it is not fully true that there is no quality loss, so I would suggest changing that by “without significant quality loss”.

We updated the above-mentioned lines (now line 30 and 331) as suggested by the Reviewer.

- Lines 42-43. The authors quote 1 μm emittances at 1 nC and 0.2 μm emittances for 10 pC and cite the work done at SwissFEL (Ref. 3). However, the quoted numbers in Ref. 3 are 200 nm for 200 pC and 100 nm for 10 pC. The authors should correct that.

We thank the Reviewer for pointing this out. We corrected the sentence to “... of the order of $\varepsilon_n \approx 1 \mu\text{m-rad}$ for high charge (~ 1 nC) and $\varepsilon_n \approx 0.1 \mu\text{m-rad}$ for low charge (~ 10 pC) operation, respectively.”.

- When discussing that the PWFA can be driven by conventional linacs or by LWFA, the authors could cite the very recent demonstration on LWFA-PWFA (<https://journals.aps.org/prx/abstract/10.1103/PhysRevX.12.041016>).

Many thanks for the suggestion. We are of course very aware of the further encouraging results in this area, and since that paper is now officially published, we are gladly adding this in line 89 as Foerster, F. M. et al. Phys. Rev. X 12, 041016 (2022). Note that we also added the recently published seeded FEL results by Labat, M. et al. Nat. Photonics 1-7, (2022) in (line 75) “... demonstration of FEL gain in the EUV²⁸ and IR²⁹ range and seeded FEL³⁰.” and in (lines 323-324) “...for free-electron-lasing in the EUV and IR range^{28,29} and seeded FEL³⁰,...” for an up-to-date citation picture.

- Line 138. I guess that the “23 (20) nm-rad” should be “23 (17) nm-rad”. This would fit with the numbers given in Table 1 of the supplementary material and with the fact that, according to the authors’ response and manuscript, the slice emittance seems to slightly increase from W1 to W2.

We thank the Reviewer for this; the numbers quoted in the main manuscript actually correspond to a position slightly downstream in the plasma stage than W1 in the table. We updated the numbers in line 138 such that it is consistent with W1 in the Supplementary Table 1. The section reads now (lines 137-138): “... and ultralow projected (average slice) emittance of $\varepsilon_{n,(x,y)} \approx 23$ (17) nm-rad.” and (lines 329-330): “...and normalized projected (average slice) emittance of 23 (17) nm-rad.”.

- Line 199: “(less than 2 nm-rad average slice emittance (...))”. The authors should check this number (according to Table 1 of the supplemental material it should be 3 nm).

We thank the Reviewer for pointing this out, too. We updated this, and it reads now (lines 199-200): “...(less than 3 nm-rad average slice emittance $\varepsilon_{n,s}$ growth)...”.

- Lines 291-306 and figure 4. The authors should comment on the fact that the temporal profile has only one mode but the spectrum shows two modes (sorry that I forgot to mention this in the first review).

Indeed, at first glance it might look like there is only one mode in the temporal profile, however, a close look reveals that also the temporal profiles contain one pronounced mode and a slightly weaker second mode. This is more clearly visible in the figure below with the logarithmic y-scale, where a) and b) show the temporal profile for the 1.5 and 0.8 Angstrom cases, respectively. Therefore, we talk about “(...) **near single-spike** radiation pulses are automatically produced (...)” in the manuscript line 292 and then in line 302 “The existence of only two wavelength modes in both cases is signature of **pronounced longitudinal coherence** (...)”. The terminology of “**near single-spike**” and “**pronounced longitudinal coherence**” indicate that it is close to a single spike regime but is not fully there, yet, which is supported by the theoretically estimated mode number of 1.6-1.7 as well.

- Table 1 of the supplementary material. From W3 to W4 there is a slight increase of the pulse duration, so I would expect a proportional decrease of the peak current. However, the current values quoted in the table are the same.

The decompression of the bunch at the chicane occurs largely at the beam tail because of the non-negligible correlated energy spread. The peak current of the witness beam is not significantly affected because the longitudinal phase space in this region is nearly flat compared to the witness beam tail. However, averaging over the entire bunch results in a slightly longer r.m.s. bunch duration. This can be seen from the figure below where we show the current profile of the witness beam at the plasma stage exit (gray dashed line) and undulator entrance (green solid line), respectively. Some charge migration occurs at the beam head and tail, however, without affecting the central part of the witness beam. To increase clarity and provide further insight we updated the sentence in line 193: “...slightly longer r.m.s. duration of ≈ 570 as due to minimal decompression of the bunch tail in the chicane, ...”.

I still think that the main idea of the paper is not new and that the results were somewhat expected from earlier work. Nevertheless, I acknowledge the advances of the authors' research. They show with high-fidelity simulations a great brightness improvement compared to previous work, the capacity to transport the electron beam without losing too much of its quality, and the capacity of such electron beams to produce GW sub-fs hard X-ray FEL pulses.

We thank the Reviewer for acknowledging our key achievements.

To conclude, I recommend publication in Nature Communications after the authors address the comments previously mentioned.

Thank you very much.

REVIEWERS' COMMENTS

Reviewer #2 (Remarks to the Author):

Dear authors of the manuscript "Attosecond-Angstrom free-electron-laser towards the cold beam limit",

Thank you for the detailed and comprehensive response to my comments as well as the corresponding revision of the paper. I think after this nice collaboration, the paper is now in very good condition and suitable for publication.

Reviewer #3 (Remarks to the Author):

The authors have satisfactorily addressed the few remaining points and the manuscript is now ready to be published.

We would like to thank all Reviewers for their efforts and contributions to the peer-review process.

Please find the corresponding revised manuscript attached and our point-to-point response (in blue) below:

Reviewers' Comments:

Reviewer #2 (Remarks to the Author):

Dear authors of the manuscript "Attosecond-Angstrom free-electron-laser towards the cold beam limit",

Thank you for the detailed and comprehensive response to my comments as well as the corresponding revision of the paper. I think after this nice collaboration, the paper is now in very good condition and suitable for publication.

We thank the Reviewer very much and agree on the fruitful outcome of the collaborative peer-review process.

Reviewer #3 (Remarks to the Author):

The authors have satisfactorily addressed the few remaining points and the manuscript is now ready to be published.

We thank the Reviewer very much.